# Proximate Chemical Composition, Amino Acids Profile and Minerals Content of Meat Depending on Carcass Part, Sire Genotype and Sex of Meat Rabbits

**DOI:** 10.3390/ani12121537

**Published:** 2022-06-14

**Authors:** Robert Gál, David Zapletal, Petra Jakešová, Eva Straková

**Affiliations:** 1Department of Food Technology, Faculty of Technology, Tomas Bata University in Zlín, 760 01 Zlín, Czech Republic; gal@ft.utb.cz; 2Department of Animal Breeding, Animal Nutrition and Biochemistry, Faculty of Veterinary Hygiene and Ecology, University of Veterinary Sciences Brno, 612 42 Brno, Czech Republic; p.humpolcova@seznam.cz (P.J.); strakovae@vfu.cz (E.S.)

**Keywords:** meat-type rabbit, sire genotype, gender, meat quality, amino acid, mineral content

## Abstract

**Simple Summary:**

Rabbit meat is popular with consumers mainly for its high-quality protein. Among the main factors influencing the characteristics of rabbit meat proteins to appertain the breed, genotype, carcass part and age. Conventional production of rabbit meat in many European countries is mainly ensured by intensive production systems, when commercial meat-type albinotic rabbit crossbreds are used. However, spotted and solidly coloured lines of meat rabbit breeds have begun to be used in rabbit breeding schemes as some consumers have begun to negatively perceive meat from albinotic coloured rabbits. The aim of the present study was to assess the effects of the sire genotype, sex and carcass part on the composition of meats of rabbits fattened under conditions where no synthetic drugs were used. Crossbreeding of Mecklenburger Schecke sires with a commercial dam line of HYLA rabbits resulted in a worse nutritional quality of meat proteins in progeny. These findings point to a possible risk of alterations in the nutritional quality of meat proteins when using different rabbit sire genotypes than those originally intended for the specific commercial crossbreeding scheme.

**Abstract:**

The aim of the study was to assess the effects of the sire genotype, sex and carcass part on the composition of the meat of rabbits, which were fattened under conditions where no synthetic drugs were used. As for carcass parts, the higher content of both total amino acids (AA) and all essential AA (EAA) monitored was found in the *Longissimus thoracis et lumborum* (*LTL*) muscle as compared to hind leg meat (*p* ˂ 0.001). Significant effects of the rabbit sire genotype and the genotype x sex interaction on proportions of some AA in meat were found (*p* ˂ 0.001). Crossbreeding of the Mecklenburger Schecke (MS) sires with a commercial dam line of HYLA rabbits resulted in a lower proportion of the total AA and all EAA monitored in meats of MS sired males as compared to MS sired females and HYLA rabbits (*p* ˂ 0.05). The sex-related effect on AA profile was not so noticeable in final commercial crossbreds of HYLA rabbits when compared to MS sired progeny. These findings point to a possible risk of alterations in the nutritional quality of meat proteins when using different rabbit sire genotypes than those originally intended for the specific commercial crossbreeding scheme. However, on the contrary, higher contents of magnesium (*p* ˂ 0.05), manganese (*p* ˂ 0.001) and zinc (*p* ˂ 0.05) were found in meats of MS sired progeny as compared to HYLA rabbits.

## 1. Introduction

Rabbit meat shows excellent nutritional and dietetic properties; moreover, it can also be effectively fortified with bioactive compounds to provide consumers an outstanding functional food [1,2]. Rabbit meat possesses a very low content of fat and cholesterol, a high level of proteins with essential amino acids (AA), no uric acid and low purine content [3]. Furthermore, rabbit meat is low in monounsaturated fatty acids, high in n-3 polyunsaturated fatty acids and it is a significant source of vitamin B (vitamins B_2_, B_3_, B_5_, B_6_, B_12_) [4]. It is low in sodium (Na) and rich in phosphorus (P) and selenium [5,6].

Rabbit meat is often popular with consumers mainly for its high-quality protein, which shows a higher digestibility value compared to other meats such as beef or pork [7,8]. The quality of meat proteins is affected by various factors and complex interactions among the biological traits of an animal [9,10]. The main factors influencing the characteristics of rabbit meat proteins, particularly an intrinsic AA composition, are breed [8], genotype [11], carcass part [12] and age [8]. Studies focusing on the evaluation of the influence of gender on AA profile in rabbit meat are still limited, whereas no sex effect on essential amino acids (EAA) proportion in rabbit meat was reported earlier by [13]. 

Conventional production of rabbit meat in many European countries is mainly ensured by intensive fattening of commercial meat-type rabbit crossbreds [14]. In addition to high growth rates and favourable feed conversions, these broiler rabbits also show high carcass value and meat quality, whereas the issue is that mostly albinotic hybrid genotypes are often reared for meat production. Recently; however, spotted- and solidly-coloured lines of meat rabbit breeds have begun to be used in rabbit breeding schemes as some consumers have begun to negatively perceive meat from albinotic coloured rabbits, which allegedly evoke laboratory-bred experimental rabbits [15]. A potential of coloured rabbit breeds and genotypes for meat production was demonstrated recently in some studies [16,17,18]. Due to their typical production traits, a lot of these rabbit genotypes can be included into specific crossbreeding schemes and utilised in the European organic and alternative production systems [17,19,20].

We found in our previous study that the use of the Mecklenburger Schecke (MS) males as terminal sires in a rabbit crossbreeding scheme led to favourable growth performance and some of the carcass characteristics in their progeny. The MS breed is medium-sized, and these rabbits show a well-muscled cylindrical body, with wide fore- and hindquarters. The breed displays three typical colour genotypes, which also differ in growth performance and meatiness. In this regard, solidly- (genotype *kk)* and spotted- (genotype *Kk)* coloured rabbits excel in important production traits. When they are crossed with white-coloured rabbits, they pass a spotted or solidly dark colour of coat on their progeny [15]. 

As the effect of inclusion of the MS breed in a crossbreeding scheme on the meat composition of their progeny has not yet been examined, the aim of the present study was to assess the effect of crossing of MS sires with the commercial dam line of HYLA rabbits on meat composition in their progeny fattened under intensive production systems where no synthetic veterinary and anticoccidial drugs were used. Further, an integral part of the present study was to evaluate the effects of sex and carcass part on the composition of rabbit meat. 

## 2. Materials and Methods

### 2.1. Animals and Management

The study was performed in the commercial Centre of HYLA rabbits (Jaroměřice nad Rokytnou–Ratibořice, Czech Republic). On the farm, which focuses mainly on the production of breeding HYLA crossbred rabbits used in other commercial farms, neither synthetic allopathic veterinary drugs nor synthetic anticoccidial drugs are used. A total of 112 crossbred rabbits (control and experimental groups) were used in the experiment. The control group (H; 28 males + 28 females) consisted of albino crossbred rabbits of the HYLA combination AB sires (*n* = 7 sires) × CD dams. The experimental group (28 males + 28 females) consisted of MS sired progeny, when the semen of MS males was inseminated to females of the same HYLA CD line as in the H group. The used MS sires (*n* = 6 sires) originated from small-scale hobby breeding stocks; all of them were *kk* or *Kk* genotypes. Females of the HYLA CD line were inseminated with the heterospermic insemination dose, which contained spermatozoa of all involved MS sires. 

### 2.2. Growing of Rabbits

The experimental design of the study was conducted according to [21], which respected guidelines for experiments with rabbits reared for meat. During the whole experimental period, rabbits of both groups were raised and fattened under identical management conditions. After weaning (35 days of age), rabbits were housed in wire cages (2 rabbits per cage) with the floor density of 0.18 m^2^ per rabbit. The cage size was 90 cm × 40 cm × 35 cm (length × width × height). The lighting period was 12 h light/12 h darkness, temperature ranged from 17 °C to 20 °C and relative humidity was 55% to 60%. Rabbits were fed ad libitum by commercial compound pelleted feeds (De Heus a.s., Běstovice, Czech Republic) and the grower (K-Optimum) and finisher (K-Finisher) diets were used from the 35th to 64th day of age and from the 65th day of age to slaughter, respectively. The nutrient composition of the diets used is shown in Table 1; the grower diet contained an anticoccidial agent Emanox which is an extract of aromatic plants.

### 2.3. Slaughter of Rabbits and Meat Samples

At the end of fattening (108 days of age), 24 rabbits per genotype (12 males and 12 females), were randomly selected (one rabbit per cage), weighed and slaughtered after a previous 12 h fasting in the abattoir. The rabbits were mechanically stunned with a captive bolt gun and bled, then the skin, distal parts of the tail, gastrointestinal and urogenital tracts and the distal part of the legs were removed according to the methodology described by Blasco and Ouhayoun [22] for rabbit meat research. Thereafter rabbit carcasses were placed in a cold storage chamber at a temperature of four °C. After 24 h, from a chilled carcass, both *LTL* muscles and hind legs were dissected. Then, both hind legs were deboned. Finally, samples of the *LTL* and hind leg meat were packed and stored at −20 °C until analysed.

### 2.4. Laboratory Methods

All samples were analysed in triplicate. The dry matter (DM) of the samples was determined by weight upon drying the sample at 105 °C under the prescribed conditions. The Kjeldahl method using a Buchi analyser (Centec Automatika, spol. s.r.o., Prague, Czech Republic) was performed to determine the crude protein (CP) content. A Soxhlet method was used to determine the ether extract by a Soxtec apparatus (Thermo Stientific, Warrington, UK). The ash was determined by weighing the sample after incineration at 550 °C. Water/protein (W/P) ratio was calculated from the formula: (1000–dry matter content)/CP content.

AA contents were determined following acid hydrolysis in 6 N HCl at 110 °C for 24 h using the Automatic Amino acid Analyzer AAA 400 (Ingos a.s., Prague, Czech Republic), based on the colour-forming reaction of AA with the oxidative agent ninhydrin according to procedures used by Straková et al. [23]. The AA analysis was used to determine the value of pure protein, expressed as the sum of EAA–lysine (Lys), leucine (Leu), isoleucine (Ile), threonine (Thr), arginine (Arg), histidine (His), phenylalanine (Phe), valine (Val), methionine (Met) and of non-essential AA (NEAA)–serine (Ser), asparagine (Asp), glutamine (Glu), proline (Pro), glycine (Gly), alanine (Ala), tyrosine (Tyr). Because it was not possible to validly determine the cysteine content due to the used AA analysis, and the tryptophan content was not determined for the objective assessment of the quality of rabbit meat protein either, the representation of individual AAs assessed as their proportion from the total CP content in the analysed meat is stated in this work. The levels of mineral elements were determined through incinerating and leaching the sample by extraction and the subsequent titration according to the Association of Official Agricultural Chemists [24]. The content of the potassium (K), sodium (Na), calcium (Ca), magnesium (Mg), copper (Cu), iron (Fe), manganese (Mn), and zinc (Zn) was determined using an atomic absorption spectrometer Agilent Technologies 200 Series AA (HPST, s.r.o., Prague, Czech Republic). The phosphorus (P) content was determined using a spectrophotometer Helios α (Thermo Scientific, Great Britain).

### 2.5. Statistical Analysis

The arithmetic mean and standard error of the mean (SEM) were determined for all assessed traits in respective evaluated groups. A Shapiro-Wilk test was used to test the normality of data distribution within the assessed groups. The normality was found in all the variables. Statistical evaluation of data followed basic procedures [25]. Differences in proximate chemical composition, AA proportion and mineral elements content between assessed carcass parts were carried out by ANOVA, with carcass part as a fixed effect and a random term for cage. Regarding assessment of genotype and sex effects, general linear model (GLM) procedure was used, where genotype and sex were included as fixed effects and their interaction as an interaction term. Random terms included slaughter weight (SW) and cage. Differences among groups were tested by Tukey’s post-hoc test. Significance was considered at the level *p* ˂ 0.05. All statistical procedures were performed by the STATISTICA CZ version 10 software.

## 3. Results

### 3.1. Slaughter Traits and Proximate Chemical Composition of Meat

The higher dry matter content of rabbit *LTL* was related to the higher CP content and lower W/P ratio (*p* ˂ 0.001) as compared to hind leg meat (Table 2). The hind leg meat displayed the higher content of ether extract and ash than the *LTL* muscles (*p* ˂ 0.001). 

When compared to the control (final crossbreds of H rabbits), a higher slaughter weight of MS sired progeny at 108 days of age (*p* ˂ 0.001; Table 3) was associated with a higher weight of carcass (*p* ˂ 0.001), hind leg meat and *LTL* muscles (*p* ˂ 0.05). On the contrary, H rabbits displayed a higher hind legs yield as compared to the MS sired rabbits (*p* ˂ 0.001). The sex of rabbits affected the carcass dressing and a higher level was found in males compared to females (*p* ˂ 0.05). Concerning basic indicators of the proximate chemical composition of meat in both assessed carcass parts, the rabbit genotype influenced only the ash content in the hind leg meat; the higher value of ash content was observed in H rabbits (*p* ˂ 0.01). The sex of rabbits affected only the content of ether extract in *LTL*, the lower value of ether extract was found in females compared to males (*p* ˂ 0.05). In addition, a significant effect of genotype x sex interaction on the content of CP and ash (*p* ˂ 0.001) and on the W/P ratio (*p* ˂ 0.01) in hind leg meat was found.

### 3.2. Amino Acids Profile

As for carcass parts assessed (Table 2), the higher content of both total AA and all EAA (*p* ˂ 0.001) monitored was found in the *LTL* muscle as compared to hind leg meat. In particular, a higher proportion of Leu, Iso, Thr, Arg, His, Val, Met, Ser, Asp and Tyr was found in the CP of *LTL* compared to the CP of hind leg meat (*p* ˂ 0.001). By contrast, higher proportions of Gly (*p* ˂ 0.001) and Phe (*p* ˂ 0.01) were found in the CP of hind leg meat than in *LTL* meat.

Concerning a rabbit genotype (Table 4), a higher proportion of the total AA and all NEAA monitored (*p* ˂ 0.001) as well as all EAA (*p* ˂ 0.05) was found in the CP of hind leg meat of H rabbits as compared to MS sired rabbits. Similarly, in the CP of *LTL* meat, the higher proportion of total AA (*p* ˂ 0.05) and all NEAA monitored (*p* ˂ 0.01) was found in H rabbits as compared to MS sired rabbits. As for the proportion of particular AA in the CP of hind leg meat, higher values of Leu, Iso, Thr, Pro, Gly and Tyr (*p* ˂ 0.001), of Val and Ser (*p* ˂ 0.01) and also of Asp and Glu (*p* ˂ 0.05) were found in H rabbits as compared to MS sired rabbits. Concerning the proportion of particular AA in the CP of *LTL* meat, higher values of Gly and Tyr (*p* ˂ 0.001), then of Leu, Thr, Val and Ser (*p* ˂ 0.01) and also of Iso, Asp and Glu (*p* ˂ 0.05) were observed in H rabbits than in MS sired rabbits.

Regarding the effect of gender (Table 4), in the CP of hind leg meat, a higher proportion of all EAA (*p* ˂ 0.01) and the total AA monitored (*p* ˂ 0.05), in particular of Phe (*p* ˂ 0.001) and also of Lys, His and Ala (*p* ˂ 0.01) was found in females as compared to males. By contrast, there was a higher proportion of Met and Tyr (*p* ˂ 0.01), as well as Gly (*p* ˂ 0.05) in the CP of male hind leg meat as compared to female hind leg meat. In addition, a significant effect of genotype x sex interaction on the proportion of all EAA (*p* ˂ 0.001), total AA (*p* ˂ 0.01), as well as NEAA (*p* ˂ 0.05) monitored in CP of hind leg rabbit meat was found; specifically, on the proportion of Lys, His, Phe, Tyr, Glu, Val, Asp and Gly. Within *LTL*, the rabbit sex influenced the proportion of all EAA monitored (*p* ˂ 0.05), of Tyr (*p* ˂ 0.001) and of Ala, Lys and Arg (*p* ˂ 0.05) in CP of this meat; the higher proportion of total EAA, Ala, Lys and Arg and the lower Tyr proportion was found in females. Moreover, the rabbit genotype x sex interaction had a considerable effect on the proportion of the total AA and all EAA (*p* ˂ 0.001) and of all NEAA monitored (*p* ˂ 0.01) in *LTL* meat; specifically, on the proportion of Arg, His, Phe, Tyr, Lys, Val, Asp, Glu, Gly, Thr and Ser. 

### 3.3. Mineral Elements

As for rabbit carcass parts, the higher Ca level (*p* ˂ 0.01) and lower K and Cu (*p* ˂ 0.01) and Zn (*p* ˂ 0.05) level was observed in hind leg meat as compared to *LTL* meat (Table 2). 

Regarding a rabbit genotype (Table 5), the higher content of Mg and Mn (*p* ˂ 0.001), Zn (*p* ˂ 0.01) and K (*p* ˂ 0.05) in hind leg meat was found in MS sired rabbits as compared to H rabbits. In *LTL* meat, a higher level of Na and Mn (*p* ˂ 0.001), Mg and Zn (*p* ˂ 0.05) was observed in MS sired rabbits than in H rabbits. 

Concerning a sex effect (Table 5), a higher level of Ca, Fe and Zn (*p* ˂ 0.05) and lower level of Mg (*p* ˂ 0.01) was found in the hind leg meat of females as compared to males. A significant effect of genotype x sex interaction on the content of Na (*p* ˂ 0.001), Ca and Fe (*p* ˂ 0.01) in this meat was also found. In *LTL* meat, a higher content of K was found in males compared to females (*p* ˂ 0.05). In addition, the rabbit genotype x sex interaction had a significant effect on the content of Fe (*p* ˂ 0.05) in *LTL*. 

## 4. Discussion

### 4.1. Slaughter Traits and Proximate Chemical Composition of Meat

A higher SW of MS sired progeny at 108 days in the present study is similar to that found in 100-day-old New Zealand White rabbit males [26] and considerably higher than the 112-day-old Burgundy Fawn and Vienna Blue sired crossbred rabbits reared under organic production [17]. The value of SW in HYLA rabbits of the present study was significantly lower than in the same rabbit genotype fattened for 103 days [27]. Although MS sired progeny of the present study attained higher SW during the extended fattening which was also related to a heavier weight of hind leg and *LTL* meat, a higher yield of hind leg meat was found in fattened H rabbits than in MS sired rabbit progeny (13.5 vs. 13.1% of SW, respectively). This fact is demonstrated by the higher muscularity of hind legs in final commercial crossbred H rabbits. The hind leg meat yield of rabbit genotypes in the present study was similar to that reported for the Belgian Giant Grey and Termond White rabbits [28]. In the case of *LTL* muscles, their yield was also slightly higher in H rabbits than in MS sired rabbits (7.8 vs. 7.5% of SW, respectively) and a higher *LTL* yield was found in H females than in MS sired females (7.8 vs. 7.3%, resp.). In the present study, the lower meat yield of the assessed rabbit groups was found in MS sired female progeny. Additionally, slightly lower values for the *LTL* yield (6.9 to 7.1%) were found in 78-day-old rabbits [16], as compared to 108-old rabbits of the present study; however, it is well known that the yield of hind leg meat increases with rabbit age [29]. Within the rabbit commercial breeding scheme, breeding for increased muscle volume in distinctive lines has begun recently. Due to the interest of consumers, the goal of this selection was focused mainly on increasing the volume of hind leg meat [30,31]. This is probably related to the findings of the present study in which a higher meat yield was found in final commercial crossbred H rabbits. In addition, it was found that a rabbit line selected for a higher thigh muscle volume also displayed some better production traits during fattening [32]. 

Regarding the proximate chemical composition of meat in the present study, the assessed traits were influenced mainly by a carcass part and the higher DM content of the *LTL* muscle was associated with the higher CP content and lower W/P ratio as compared to hind leg meat. This finding is in agreement with findings reported in three-month-old rabbits [29,33]. By contrast, Króliczewska et al. found the higher CP content in hind leg meat compared to *LTL*, furthermore the DM content was also higher in *LTL* meat of the five and a half-month-old New Zealand White rabbits [34]. The lower W/P ratio in *LTL* of the present study favours this rabbit meat for subsequent meat processing, since it contributes to a higher product yield due to lower loss of its own water and better ability to hold added water [33]. When compared to *LTL*, a higher intramuscular fat (IMF) level was found in hind legs in the present study, furthermore its content is inversely associated with the content of CP and water in rabbit meat [35]. This finding is in agreement with earlier reported results [29,33]; however, the IMF value in hind legs of the present study is still generally low, which confirms a leanness of rabbit meat [36]. Higher ash content was found in hind leg meat in the present study as compared to *LTL*. By contrast, the opposite trend for the ash level was found in the Pannon White rabbits [29]; their SW were similar to rabbits in the present study. The total mineral content in meat in the present study is similar to that found by Perna et al. and by Daszkiewicz and Gugolek [26,33].

A rabbit genotype in the present study influenced only the total mineral content in hind leg meat, its higher value was found in H rabbits. Metzger et al. [29] mention that rabbits with higher body weight (BW) at a given age may have a lower ash content in hind leg meat, which is in agreement with the finding of the present study in which MS sired females gained the higher SW and the significantly lowest total mineral content.

In the present study, a rabbit’s gender affected only the IMF level in *LTL*; leaner meat was found in females. Similarly, Ortiz Hernández and Rubio Lozano reported lower IMF content in 70-day-old females of the New Zealand White breed compared to their males; nevertheless, they found the opposite trend in the Californian breed [37]. It is known that a higher growth intensity of rabbit results in a higher IMF content, if measured at the same age [29]. In the present study, rabbit SW considerably differed between genotypes assessed, which is a reason why the SW was used as a random effect in the statistical equation. Therefore, the slightly higher IMF content in hind leg meat of MS sired rabbits was not different from that found in H rabbits. However, the IMF content of *LTL* meat was higher in males of the present study, which also was observed (*p* ˃ 0.05) by North et al. [38] in the *LTL* of meat rabbit genotypes. It seems that male rabbits may in some cases deposit more IMF into the *LTL* muscle with advancing age. Based on the IMF content of hind leg meat in the present study, MS sired females were most mature at the given slaughter age.

### 4.2. Amino Acid Profile

In general, meat AA composition is influenced by different syntheses of AA as related to different biological stages of animals [8]. For consumers it is important to meet demands for EAA, as the intrinsic contents of EAA are usually used to assess a biological value of proteins [39]. A better nutritional value of proteins in *LTL* as compared to hind leg meat in rabbits of the present study was presented by a higher proportion of total AA (+2.9%) and particularly all EAA assessed (+11.4%). On the other hand, in hind leg meat, a higher proportion of Phe (+9.2%) and Gly (+7.9%) was found. In particular, the higher proportion of seven out of nine EAA assessed and of three NEAA was found in *LTL* meat. The order of representation of individual EAAs was almost identical in both evaluated rabbit carcass parts, with the following order starting from the highest proportion: Lys → Leu → Arg → Val → Iso. Regarding NEAA, the order of the individual AAs was again almost identical in both evaluated carcass parts, the highest proportions were demonstrated for Glu → Asp → Ala → Gly. 

The above stated findings of the present study are in agreement with those found by Migdal et al. and Bivolarski et al. in the three-month-old New Zealand White rabbits [12,40] and by Nasr et al. in *LTL* of 70-day-old rabbits of different genotypes [11] and also by Liu et al. in *LTL* of the 105-day-old Ira rabbits [8]. Results of the present study confirm dietetic properties of rabbit meat due to high proportion of EAA from total AA content. In addition, regarding the proportion of some EAA, the level of Leu, Arg and Thr in *LTL* meat considerably increased with an advanced rabbit age [8]. 

In the present study, a surprisingly significant effect of sire genotypes on the AA composition in both evaluated carcass parts was found. Both in *LTL* and especially in hind legs, the higher nutritional value of meat CP was found in H rabbits as compared to MS sired rabbits. In the case of hind leg meat, a higher content of both total AA (+5.9%) and EAA monitored (+5.3%) was found in H rabbits than the MS sired rabbits and a higher proportion of four out of nine EAA and higher proportion of six out of seven NEAA monitored was found in the CP of hind leg meat in H rabbits. However, the lowest values of these EAA and three out of 7 NEAA displayed the MS sired males, furthermore the MS sired females did not show such a decrease in quality of hind leg proteins when compared to H rabbits. In particular, the most significant variation in this meat was found for Tyr (27.5%), Thr (12.3%), Leu (10.8%), Val (10.3%) and Iso (10.2%). As for *LTL*, similarly higher content of total AA (+4.3%) was found in H rabbits than in MS sired rabbits and a higher proportion of four out of nine EAA and a higher proportion of five out of seven NEAA monitored was found in the CP of *LTL* meat in H rabbits. When compared to MS sired progeny, the largest difference in the content of Tyr (+23.5%) and in the content of Gly (+13.6%) and Val (+9.2%) was found again in the CP of *LTL* meat in H rabbits. A genotype effect on the AA profile in *LTL* rabbit meat was recently observed as well [8,11]. Among the nine rabbit genotypes assessed [11], it was found that the genotype influenced proportions of two out of ten EAA and three out of seven NEAA; the largest variations were found in the proportion of cysteine (88%), His (31%) and Phe (22%). Li et al. [8] found differences in the content of three EAA and 2 NEAA between the two rabbit breeds, with considerable variation in the Tyr level that is similar to that found in the present study. Additionally, it was [41] found that differences in AA metabolism were also linked to rabbit gut microbial function capacities and they greatly varied between the meat-type breeds. Ye et al. [41,42] state that extended knowledge about host-gut microbiome metabolome interaction could aid future improvement in important production traits as well as the health of rabbits. Findings of the present study confirm high nutritional value of meat proteins in final commercial crossbreds of H rabbits, which are a result of the intended combination crossbreeding among deliberately selected rabbit lines for desired production traits. On the other hand, the findings of the present study point to a possible risk of deteriorating the nutritional quality of meat proteins when using different rabbit sire genotypes than those intended for meat production in the distinctive commercial crossbreeding scheme.

Due to higher proportion of EAA in both carcass parts assessed, a higher nutritional value of meat proteins was found in females than in males in the present study and the proportion of total NEAA in the meat CP content was not influenced by a rabbit sex. The sex had a more considerable effect on the AA profile variation in hind leg meat than in *LTL* meat. In hind legs, the rabbit sex influenced proportions of four out of nine EAA (Phe, Lys and His were higher in females and Met was higher in males) and of three out of seven NEAA (Tyr and Gly were higher in males and Ala was higher in females). In *LTL*, the sex affected proportions of 2 EAA (Lys and Arg were higher in females) and of two NEAA (Tyr was higher in males and Ala was higher in females). However, it is necessary to note that generally higher proportions of a majority of above-mentioned AA in females were demonstrated because of lower proportions of these AA in MS sired males. As demonstrated, the rabbit genotype x sex interaction played the very important role for these traits in the present study.

In contrast to the findings of the present study, no sex effect on the EAA profile in rabbit hind leg meat was observed [13], who evaluated the influence of various rabbit production systems in her study.

Furthermore, no sex differences in AA profile of meat were reported recently: neither in pigs [43], cattle [44] or horses [45]. Additionally, because the dietary EAA digestibility may differ from the protein digestibility as such, a score based on individual EAA digestibility is determined currently as a new protein quality assessment [46]. 

The taste of food is one of the main factors determining food preference and eating habits [47]; simultaneously, intrinsic AA can be also important contributors to the specific taste of many foods, including meat [48,49]. In this regard, taste-active AA can be contributors to the sweet flavour (Gly, Ala, Ser, Thr, Pro), bitter flavour (His, Arg, Ile, Leu, Lys, Phe, Val), sour taste (Phe, Tyr, Ala) and umami taste (Glu and Asp) of meat [46]. However, some taste-active AA have more than only one taste characteristic. For example, Arg has a bitter and slightly sweet sensation, Ser is sweet with some sour and umami taste, Glu (a savoury AA) has a combination of sour and umami taste and Ala has a sweet and slightly umami taste [46,48,50]. On the basis of the found genotype- and sex-related differences in AA levels in rabbit meat of the present study, a certain variation in the flavour of meat can also be expected, since the intramuscular fat which is mainly linked to a meat flavour [51,52] is generally low in rabbits. These variations in the sensation of meat flavour can occur particularly after the processing of specific meat products. 

### 4.3. Mineral Elements

A large variation in the content of minerals in rabbit meat was reported in recent studies. 

In the present study, the higher Ca content and lower K, Cu and Zn content was found in rabbit hind leg meat as compared to *LTL* meat. Furthermore, Ca content of hind leg meat differed between rabbit sexes, mainly due to its considerably higher level in H females. Calcium helps develop and maintain strong bones and teeth, increases the utilization of other minerals (i.e., P, K) and its adequate intake contributes to the prevention of cardiovascular diseases [53,54]. The Ca content in *LTL* meat of the present study is similar to that reported [54]. However, D´Arco et al. found a markedly higher Ca content (48 mg/100 g of meat) in whole edible meat gained from carcasses of 80-day-old rabbits [55].

Although a lower ash content in hind leg meat was found in MS sired rabbits in the present study, the content of four out of nine assessed minerals in this meat was higher than in H rabbits. Similarly, a higher content of 4 monitored minerals was also found in the LTL of the MS sired genotype. Thus, a higher content of natural forms of some minerals, mainly of Mg, Mn and Zn was observed in the meat of the MS sired progeny. Regarding K level, a genotype affected its value only in hind leg meat, the higher level of K was found in the MS sired progeny. Moreover, in *LTL,* a higher level was found in males than in females. In general, rabbit meat has a higher K concentration than the meat from other animal species [4]. Potassium is the most abundant mineral in rabbit meat [5,56], which is in agreement with the results of the present study. However, K concentration in meat of the present study is somewhat lower than those reported [5,56]. A high K and low Na level can make rabbit meat mainly recommended for hypertension diets, especially in sodium-sensitive individuals [5]. When compared to H rabbits in the present study, the dietetically favourable lower Na level was observed in MS sired progeny in hind leg meat, whereas contrarily, a markedly higher Na level in *LTL* was found in the MS sired genotype. The levels of Na in meat in the present study are similar to those observed in [5]. These authors found that the level of Na in meat also decreased with the increasing age of rabbits. In addition, a somewhat lower Na concentration in rabbit meat was found [54,56]. Rabbit meat is also highly recommended for its Mg content [56], through Mg’s role in enzyme activation; this mineral stimulates muscle and nerve contraction, it also plays an important role in many other metabolic functions in humans [54]. Apart from the genotype-related effect on Mg level in the present study, a higher concentration of Mg in males in the hind leg meat was also observed. Values of Mg in meat in the present study are similar to those reported [55,56]. In addition, a slight increase in the level of Mg in meat with the increasing rabbit age was found [5]. The second most abundant mineral in meat is P and rabbit meat is rich in it [56]. The level of P in rabbit meat in the present study was similar to that observed [5,55,56].

Regarding trace elements in meat of the present study, Fe was the most abundant microelement, followed by Zn, which is in agreement with the finding [54] observed in rabbits fed a diet supplemented with olive leaves. As other white meats, rabbit meat is generally low in both Fe and Zn [5,57]. Values of Fe and Zn in rabbit meat in the present study were similar to those published [52]. In addition, [58] found the considerably higher Fe content in meat (2.15 mg/100 g) of local rabbit breeds reared under an extensive production system. Even though meat in general is the main dietary source of the highly available Fe, it is important to take into consideration the particular amounts of heme and non-heme Fe [56]. The heme Fe in meat has the advantage of being more biologically available than the non-heme Fe from plant-based products; heme Fe in rabbit meat usually varies from 56 to 62% [59]. Unlike the level of Fe in the present study, the level of Zn was influenced markedly by a sire genotype and a higher content of Zn was found in MS sired rabbits in both carcass parts assessed. In addition to many other physiological functions, Zn and Fe are involved in the functioning of the antioxidant defense system as they are cofactors of certain enzymes active against free radicals [60]. In this regard, a variation in dietary Zn level is often linked to a different modulation of the immune system of organisms [61]. Similar to Zn content, a sire genotype also considerably affected Mn content, showing its twice-fold higher level in MS sired progeny than in H rabbits. Furthermore, the level of Mn in meat of H rabbits was similar to that reported [56] and markedly higher than those observed [5,55] in rabbit meat (0.03 mg and 0.02 mg/100 g, respectively). Manganese is both an essential trace element and a potential neurotoxicant [62]. An adequate Mn supply is thought to be necessary for many physiological processes and its low dietary intake was related to detrimental health effects in animals. In addition, it has recently been reported that there is no indication that Mn dietary exposure (not from drinking water) is associated with adverse health effects in humans [62,63]. The Cu level in rabbit meat does not differ considerably from its level in meat of other animal species [64]. The content of Cu in rabbit meat in the present study was similar to the levels earlier reported [5,55]. Additionally, it was demonstrated that the content of Cu in muscle is inversely related to the muscle lipid content in cattle [65].

## 5. Conclusions

A rabbit *LTL* meat displayed better dietetic properties than hind leg meat due to its lower IMF content and higher proportion of total AA, mainly of all essential AA.

Surprisingly, substantial effects of sire genotype and genotype x sex interaction of rabbits on the AA profile of meat were found. Crossbreeding of MS sires with a commercial dam line of HYLA rabbits resulted in a worse nutritional quality of meat proteins in fattened MS sired males. These findings point to a possible risk of alterations in the nutritional quality of meat proteins when using different sire genotypes than those originally intended for the specific commercial crossbreeding scheme.

On the other hand, the content of natural forms of some minerals was higher in the meat of MS sired progeny.

## Figures and Tables

**Table 1 animals-12-01537-t001:** Chemical composition (g/kg) of the diets as-fed basis.

	Grower	Finisher
Item	(Day 35 to 64)	(After Day 65)
Crude protein	157.4	154.1
Crude fibre	125.8	128.1
Crude fat	42.1	33.6
Crude starch	151.8	150.3
Ash	68.6	63.2
Calcium	8.68	6.94
Inorganic phosphorus	7.2	6.6
Asparagine	12.5	12.6
Threonine	5.3	5.4
Serine	6.3	6.4
Glutamine	28.8	27.6
Proline	8.7	9.0
Glycine	7.1	7.1
Alanine	7.1	7.3
Valine	7.5	7.4
Methionine	1.6	1.5
Isoleucine	5.4	5.4
Leucine	9.8	9.7
Tyrosine	4.7	4.3
Phenylalanine	6.3	6.0
Histidine	4.3	4.1
Lysine	8.1	7.5
Arginine	9.6	9.0

**Table 2 animals-12-01537-t002:** Effect of the carcass part on meat composition of rabbits.

Item	Meat		*p*-Value
	Hind Leg	*LTL*	
Proximate chemical composition (g/kg of fresh meat)			
Dry matter	241.7 ± 1.65	250.7 ± 1.74	<0.001
Crude protein	215.4 ± 1.16	227.4 ± 1.11	<0.001
Ether extract	24.6 ± 0.91	16.5 ± 0.67	<0.001
Ash	12.3 ± 0.05	11.9 ± 0.06	<0.001
W/P	3.53 ± 0.025	3.30 ± 0.022	<0.001
Amino acids (g/100 g of total crude protein)			
Lysine	8.86 ± 0.193	8.88 ± 0.172	0.844
Leucine	7.29 ± 0.097	7.71 ± 0.093	<0.001
Isoleucine	4.29 ± 0.055	4.52 ± 0.056	<0.001
Threonine	4.00 ± 0.055	4.27 ± 0.052	<0.001
Arginine	5.64 ± 0.107	6.33 ± 0.182	<0.001
Histidine	3.76 ± 0.063	4.15 ± 0.063	<0.001
Phenylalanine	4.26 ± 0.138	3.90 ± 0.055	0.004
Valine	4.74 ± 0.070	4.97 ± 0.071	<0.001
Methionine	1.84 ± 0.038	2.07 ± 0.042	<0.001
∑ essential AA	44.7 ± 0.05	49.8 ± 0.54	<0.001
Serine	3.44 ± 0.043	3.67 ± 0.042	<0.001
Asparagine	8.57 ± 0.097	8.95 ± 0.105	<0.001
Glutamine	14.4 ± 0.16	14.6 ± 0.16	0.150
Proline	3.76 ± 0.050	3.70 ± 0.058	0.223
Glycine	4.63 ± 0.062	4.29 ± 0.065	<0.001
Alanine	5.38 ± 0.094	5.33 ± 0.094	0.307
Tyrosine	3.37 ± 0.079	3.54 ± 0.081	<0.001
∑ non-essential AA	43.6 ± 0.46	44.1 ± 0.472	0.095
∑ all AA	88.3 ± 0.07	90.9 ± 0.91	<0.001
Mineral elements (mg/100 g of fresh meat)			
Phosphorus	229.8 ± 4.24	224.8 ± 9.60	0.558
Potassium	232.9 ± 4.35	250.3 ± 5.85	0.005
Sodium	55.4 ± 0.78	58.3 ± 1.61	0.113
Calcium	27.8 ± 1.39	24.4 ± 1.12	0.003
Magnesium	21.9 ± 0.43	22.9 ± 0.42	0.063
Copper	0.05 ± 0.002	0.06 ± 0.002	0.003
Iron	1.01 ± 0.032	1.03 ± 0.028	0.574
Manganese	0.10 ± 0.006	0.10 ± 0.005	0.514
Zinc	0.92 ± 0.014	0.96 ± 0.015	0.019

Data are means ± standard error of the mean. ∑: sum. AA: amino acid. *LTL*: *Longissimus thoracis et lumborum.* W/P: water/protein ration.

**Table 3 animals-12-01537-t003:** Carcass traits and proximate chemical composition (g/kg of fresh meat) of rabbit meats in relation to the genotype and sex.

Item	Genotype				*p*-Value		
	H		MS × H		Gen.	Sex	Gen. × Sex
	M	F	M	F			
Slaughter weight (g)	2754 ± 81.3	2941 ± 70.2	3321 ± 63.9	3331 ± 72.3	<0.001	0.167	0.213
ADG (g)	23.9 ± 1.39	26.4 ± 1.56	32.3 ± 1.05	31.9 ± 1.11	<0.001	0.490	0.267
Carcass weight (g)	1677 ± 64.1	1735 ± 78.0	2059 ± 46.4	1983 ± 48.2	<0.001	0.715	0.421
Carcass dressing (%)	60.9 ± 0.80	58.8 ± 0.97	61.1 ± 0.63	60.0 ± 0.27	0.137	0.012	0.453
Hind legs yield (%)	33.7 ± 0.29	34.0 ± 0.17	32.7 ± 0.24	32.9 ± 0.25	<0.001	0.209	0.956
Hind legs meat (g)	378.2 ± 16.40	390.7 ± 16.86	434.8 ± 11.82	432.1 ± 11.84	0.026	0.105	0.371
*LTL* (g)	210.9 ± 10.86	228.0 ± 13.75	256.5 ± 10.68	241.5 ± 7.16	0.036	0.108	0.173
Hind leg meat							
Dry matter	237.3 ± 3.26	240.1 ± 4.16	245.0 ± 2.95	244.4 ± 2.54	0.362	0.815	0.637
Crude protein	210.0 ± 2.11 ^b^	216.5 ± 2.44 ^a,b^	221.2 ± 1.39 ^a^	213.7 ± 2.09 ^a,b^	0.241	0.704	<0.001
Ether extract	22.8 ± 2.32	22.5 ± 1.51	24.0 ± 1.64	28.9 ± 1.18	0.404	0.229	0.077
Ash	12.3 ± 0.11 ^a^	12.5 ± 0.09 ^a^	12.4 ± 0.07 ^a^	12.0 ± 0.08 ^b^	0.002	0.152	<0.001
W/P	3.64 ± 0.049 ^a^	3.52 ± 0.055 ^a,b^	3.41 ± 0.024 ^b^	3.54 ± 0.046 ^a,b^	0.238	0.824	0.005
*LTL*							
Dry matter	247.8 ± 4.05	249.5 ± 3.12	255.8 ± 2.63	249.7 ± 3.88	0.827	0.436	0.336
Crude protein	223.2 ± 2.25	227.6 ± 2.75	230.1 ± 1.10	228.7 ± 2.15	0.799	0.701	0.275
Ether extract	18.3 ± 2.08	15.1 ± 0.87	17.3 ± 1.07	15.3 ± 0.86	0.166	0.028	0.451
Ash	11.7 ± 0.18	12.0 ± 0.10	12.0 ± 0.05	11.9 ± 0.07	0.860	0.358	0.121
W/P	3.38 ± 0.044	3.30 ± 0.051	3.23 ± 0.021	3.29 ± 0.046	0.740	0.960	0.239

Data are means ± standard error of the mean. ^a,b^: In rows, means with different superscript letters differ at *p* ˂ 0.05. H: HYLA. MS: Mecklenburger Schecke. M: male. F: female. Gen.: genotype. ADG: average daily gain from 35 to 108 days. *LTL**: Longissimus thoracis et lumborum*. W/P: water/protein ratio.

**Table 4 animals-12-01537-t004:** Amino acid profile (g/100 g of total crude protein) of rabbit meats in relation to the genotype and sex.

Item	Genotype				*p*-Value		
	H		MS × H		Gen.	Sex	Gen. × Sex
	M	F	M	F			
Hind leg meat							
Lysine	8.94 ± 0.102 ^a,b^	8.80 ± 0.147 ^b,c^	7.65 ± 0.164 ^c^	10.0 ± 0.57 ^a^	0.756	0.001	<0.001
Leucine	7.69 ± 0.184	7.64 ± 0.092	6.81 ± 0.137	7.02 ± 0.216	<0.001	0.895	0.270
Isoleucine	4.49 ± 0.099	4.51 ± 0.056	3.99 ± 0.077	4.18 ± 0.126	<0.001	0.453	0.219
Threonine	4.28 ± 0.096	4.19 ± 0.058	3.77 ± 0.069	3.77 ± 0.119	<0.001	0.401	0.439
Arginine	5.88 ± 0.064	5.60 ± 0.066	5.21 ± 0.086	5.86 ± 0.393	0.991	0.294	0.052
Histidine	3.97 ± 0.059 ^a^	3.81 ± 0.076 ^a^	3.26 ± 0.065 ^b^	4.03 ± 0.144 ^a^	0.091	0.003	<0.001
Phenylalanine	4.07 ± 0.087 ^b^	3.99 ± 0.202 ^b^	3.54 ± 0.153 ^b^	5.43 ± 0.267 ^a^	0.114	<0.001	<0.001
Valine	5.07 ± 0.148 ^a^	4.87 ± 0.071 ^a^	4.35 ± 0.093 ^b^	4.66 ± 0.144	0.002	0.883	0.034
Methionine	2.05 ± 0.082	1.76 ± 0.047	1.87 ± 0.084	1.69 ± 0.043	0.444	0.004	0.477
∑ essential AA	46.4 ± 0.69 ^a^	45.2 ± 0.67 ^a^	40.5 ± 0.76 ^b^	46.7 ± 0.73 ^a^	0.034	0.003	<0.001
Serine	3.63 ± 0.069	3.57 ± 0.055	3.31 ± 0.056	3.26 ± 0.107	0.001	0.285	0.718
Asparagine	8.90 ± 0.124 ^a^	8.64 ± 0.183 ^a,b^	8.10 ± 0.163 ^b^	8.36 ± 0.231 ^a,b^	0.011	0.694	0.015
Glutamine	14.8 ± 0.20 ^a^	14.7 ± 0.23 ^a^	13.4 ± 0.28 ^b^	14.8 ± 0.37 ^a^	0.012	0.105	0.006
Proline	3.83 ± 0.079	3.99 ± 0.094	3.62 ± 0.075	3.60 ± 0.113	<0.001	0.726	0.523
Glycine	5.05 ± 0.072 ^a^	4.59 ± 0.130 ^b^	4.41 ± 0.084 ^b^	4.46 ± 0.118 ^b^	<0.001	0.022	0.008
Alanine	4.97 ± 0.259	5.73 ± 0.081	5.16 ± 0.111	5.67 ± 0.157	0.477	0.003	0.666
Tyrosine	4.14 ± 0.076 ^a^	3.42 ± 0.093 ^b^	2.84 ± 0.042 ^c^	3.09 ± 0.053 ^c^	0.000	0.001	<0.001
∑ non-essential AA	45.3 ± 0.69 ^a^	44.6 ± 0.73 ^a^	40.8 ± 0.76 ^b^	43.5 ± 0.98 ^a,b^	<0.001	0.459	0.020
∑ all AA	91.8 ± 1.37 ^a^	89.8 ± 1.36 ^a^	81.3 ± 1.51 ^b^	90.2 ± 0.87 ^a^	0.002	0.031	0.001
*LTL*							
Lysine	9.03 ± 0.088 ^a,b^	8.72 ± 0.130 ^a,b^	8.01 ± 0.187 ^b^	9.75 ± 0.558 ^a^	0.598	0.017	0.003
Leucine	7.97 ± 0.129 ^a^	7.84 ± 0.112 ^a,b^	7.24 ± 0.199 ^b^	7.78 ± 0.225 ^a,b^	0.005	0.518	0.027
Isoleucine	4.66 ± 0.086	4.60 ± 0.067	4.26 ± 0.113	4.55 ± 0.142	0.035	0.469	0.083
Threonine	4.48 ± 0.074 ^a^	4.36 ± 0.066 ^a^	3.99 ± 0.110 ^b^	4.26 ± 0.111 ^a,b^	0.009	0.645	0.036
Arginine	6.46 ± 0.060 ^a,b^	5.89 ± 0.097 ^b^	5.51 ± 0.140 ^b^	7.46 ± 0.583 ^a^	0.273	0.023	<0.001
Histidine	4.31 ± 0.058 ^a,b^	4.02 ± 0.056 ^b,c^	3.80 ± 0.097 ^c^	4.49 ± 0.160 ^a^	0.912	0.060	<0.001
Phenylalanine	4.15 ± 0.048 ^a^	3.84 ± 0.046 ^a,b^	3.52 ± 0.086 ^b^	4.11 ± 0.132 ^a^	0.165	0.150	<0.001
Valine	5.40 ± 0.066 ^a^	4.99 ± 0.083 ^a,b^	4.57 ± 0.137 ^b^	4.94 ± 0.158 ^b^	0.002	0.652	0.002
Methionine	2.14 ± 0.073	1.97 ± 0.095	2.04 ± 0.069	2.12 ± 0.098	0.806	0.647	0.215
∑ essential AA	48.6 ± 0.44 ^a,b^	46.2 ± 0.64 ^b,c^	42.9 ± 1.08 ^d^	49.5 ± 0.97 ^a^	0.209	0.028	<0.001
Serine	3.84 ± 0.046 ^a^	3.71 ± 0.060 ^a,b^	3.45 ± 0.091 ^b^	3.66 ± 0.095 ^a,b^	0.009	0.867	0.027
Asparagine	9.40 ± 0.101 ^a^	8.90 ± 0.144 ^a,b^	8.47 ± 0.241 ^b^	9.04 ± 0.246 ^a,b^	0.047	0.934	0.009
Glutamine	15.5 ± 0.13 ^a^	14.4 ± 0.22 ^a,b^	13.8 ± 0.34 ^b^	14.6 ± 0.38 ^a,b^	0.012	0.422	0.002
Proline	3.65 ± 0.064	3.83 ± 0.106	3.60 ± 0.075	3.73 ± 0.183	0.271	0.285	0.943
Glycine	4.76 ± 0.052 ^a^	4.37 ± 0.077 ^b^	3.91 ± 0.130 ^c^	4.13 ± 0.108 ^b,c^	<0.001	0.231	0.002
Alanine	4.92 ± 0.263	5.63 ± 0.077	5.22 ± 0.147	5.57 ± 0.155	0.532	0.019	0.506
Tyrosine	4.34 ± 0.051 ^a^	3.49 ± 0.068 ^b^	3.09 ± 0.054 ^c^	3.25 ± 0.130 ^b,c^	<0.001	<0.001	<0.001
∑ non-essential AA	46.4 ± 0.49 ^a^	44.3 ± 0.62 ^a,b^	41.6 ± 1.06 ^b^	44.0 ± 1.01 ^a,b^	0.005	0.894	0.009
∑ all AA	95.0 ± 0.92 ^a^	90.6 ± 1.18 ^a^	84.5 ± 2.14 ^b^	93.5 ± 1.19 ^a^	0.018	0.227	<0.001

Data are means ± standard error of the mean. ^a,b,c^: In rows, means with different superscript letters differ at *p* ˂ 0.05. H: HYLA. MS: Mecklenburger Schecke. M: male. F: female. Gen.: genotype. ∑: sum. AA: amino acid. *LTL*: *Longissimus thoracis et lumborum*.

**Table 5 animals-12-01537-t005:** Content of mineral elements (mg/100 g of fresh meat) of rabbit meats in relation to the genotype and sex.

Item	Genotype				*p*-Value		
	H		MS × H		Gen.	Sex	Gen. × Sex
	M	F	M	F			
Hind leg meat							
Phosphorus	231.8 ± 9.94	227.4 ± 11.57	228.7 ± 5.44	231.0 ± 6.62	0.218	0.605	0.462
Potassium	243.8 ± 12.33	212.5 ± 6.15	240.5 ± 4.78	235.1 ± 7.55	0.014	0.129	0.328
Sodium	60.1 ± 1.17 ^a^	52.9 ± 1.40 ^b^	52.2 ± 1.22 ^b^	56.5 ± 1.41 ^a,b^	0.238	0.271	<0.001
Calcium	25.0 ± 1.48 ^b^	37.0 ± 4.36 ^a^	26.1 ± 0.73 ^b^	23.1 ± 0.64 ^b^	0.341	0.038	0.002
Magnesium	20.5 ± 0.61	19.3 ± 0.46	25.3 ± 0.80	22.7 ± 0.35	<0.001	0.008	0.249
Copper	0.04 ± 0.003	0.05 ± 0.004	0.06 ± 0.002	0.06 ± 0.002	0.506	0.243	0.090
Iron	0.87 ± 0.071 ^b^	1.16 ± 0.090 ^a^	1.02 ± 0.013 ^a,b^	1.00 ± 0.013 ^a,b^	0.847	0.023	0.009
Manganese	0.06 ± 0.009	0.07 ± 0.011	0.14 ± 0.004	0.13 ± 0.004	<0.001	0.747	0.485
Zinc	0.87 ± 0.026	0.88 ± 0.017	0.92 ± 0.029	1.02 ± 0.013	0.003	0.016	0.094
*LTL*							
Phosphorus	219.4 ± 9.00	228.9 ± 16.75	228.1 ± 7.01	223.0 ± 5.46	0.172	0.735	0.967
Potassium	272.2 ± 12.30	238.0 ± 14.66	260.3 ± 5.55	230.7 ± 9.33	0.773	0.018	0.969
Sodium	52.1 ± 1.26	48.5 ± 1.62	63.5 ± 2.86	69.0 ± 2.63	<0.001	0.556	0.097
Calcium	24.9 ± 1.49	23.6 ± 2.56	25.2 ± 0.47	23.9 ± 1.09	0.415	0.607	0.846
Magnesium	22.0 ± 0.69	21.1 ± 0.57	24.4. ± 0.79	24.0 ± 0.96	0.017	0.568	0.860
Copper	0.05 ± 0.003	0.06 ± 0.004	0.06 ± 0.002	0.06 ± 0.002	0.785	0.187	0.172
Iron	0.95 ± 0.07	1.14 ± 0.070	1.08 ± 0.019	0.96 ± 0.028	0.182	0.774	0.017
Manganese	0.07 ± 0.008	0.08 ± 0.008	0.14 ± 0.004	0.12 ± 0.004	<0.001	0.449	0.187
Zinc	0.93 ± 0.036	0.92 ± 0.032	1.02 ± 0.021	0.99 ± 0.026	0.015	0.643	0.616

Data are means ± standard error of the mean. ^a,b^: In rows, means with different superscript letters differ at *p* ˂ 0.05. H: HYLA. MS: Mecklenburger Schecke. M: male. F: female. Gen.: genotype. *LTL*: *Longissimus thoracis et lumborum*.

## Data Availability

Data available on reasonable request.

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
