# Peer review of "Proximate Chemical Composition, Amino Acids Profile and Minerals Content of Meat Depending on Carcass Part, Sire Genotype and Sex of Meat Rabbits"

_animals, 2022, doi:10.3390/ani12121537_

Round 1

Reviewer 1 Report

The aim of the present study was to assess the effects of the sire genotype, sex and carcass part on quality of meats of rabbits fattened under conditions where no synthetic drugs were used. The experiment design and the aims are good, but there are some problems based on this manuscript:

Throughout manuscript: the quality of English language needs to be improved.

L 24-26: Very long and confusing sentence. Revise please. “... and hind leg meats of rabbits, which fattened under conditions that no synthetic drugs were used” maybe better.

L 39-41: This sentence has a language problem, it is recommended to modify it

L 42: “MUFA” and “PUFA”, please add full name of the abbreviation

Table 1: Why no diet ingredient composition?

L 117, L 279, L 286, L 308, L 380: The author's name should be added before the reference number [21], [35], [25], [7], [52]

L 130: AA, the abbreviation can be used directly here, because the abbreviation has already appeared at L41. L 134: EAA the same.

L 282: What is “given age”?

L 304-306: The reference in this sentence is marked in an inappropriate position. In addition, the meaning of this sentence is unclear, please revise.

L 318: “and Iso”, not “a Iso”

L 359: “when” should be “simultaneously”?

L 360: ...can be several contributors for...

L 368: “that is” should be “which”?

L 438-440: Very long and confusing sentence. Revise please.

Conclusion: The conclusion is too long, please shorten it

Author Response

At first we would like to thank to the reviewer 1 for the factual comments and proposed corrections to improve the quality of our paper. In the submitted revised manuscript we have taken into consideration all the comments and suggestions.

To specific points:

L 24-26: Very long and confusing sentence. Revise please. “... and hind leg meats of rabbits, which fattened under conditions that no synthetic drugs were used” maybe better.

We corrected it.

L 39-41: This sentence has a language problem, it is recommended to modify it.

We modified it.

L 42: “MUFA” and “PUFA”, please add full name of the abbreviation.

We wrote the full names. 

Table 1: Why no diet ingredient composition?

There were used the commercial pelleted feeds in the present study. The intrinsic ingredient compositions of used diets are the proprietary by the manufacturing company (De Heus a.s.), so it is not possible to obtain and publish these data.

L 117, L 279, L 286, L 308, L 380: The author's name should be added before the reference number [21], [35], [25], [7], [52]

We corrected it.

L 130: AA, the abbreviation can be used directly here, because the abbreviation has already appeared at L41. L 134: EAA the same.

We corrected it.

L 282: What is “given age”?

“a same-age”; we corrected it in the manuscript.

L 304-306: The reference in this sentence is marked in an inappropriate position. In addition, the meaning of this sentence is unclear, please revise.

We corrected it.

L 318: “and Iso”, not “a Iso”

We corrected it.

L 359: “when” should be “simultaneously”?

We corrected it.

L 360: ...can be several contributors for...

We corrected it.

L 368: “that is” should be “which”?

We corrected it.

L 438-440: Very long and confusing sentence. Revise please.

We corrected it.

Conclusion: The conclusion is too long, please shorten it.

We shortened it; as recommended it also by the reviewer 5.

Reviewer 2 Report

The study aimed to study the effects of breed, body part and sex-related effects on meat composition of rabbits. The subject is interesting  and bring some new useful information. In addition, the manuscript is well-written and structured. I've just made few comments/suggestions (attached file).

Author Response

At first we would like to thank to the reviewer 2 for the factual comments and proposed corrections to improve the quality of our paper. In the revised manuscript we have taken into consideration all the comments and suggestions.

To specific points:

L 155: Regarding the assessment of the effect of carcass part on evaluated traits, there were not included the genotype and sex factors in this ANOVA analysis.

L 75: we corrected it.

L 158: we defined the short here.

Table 1: we deleted the column “Units”, as recommended also by other referee.

L 244: we changed to “lower”.

Reviewer 3 Report

The manuscript entitled „Proximate chemical composition, amino acids profile and minerals content of meat depending on carcass part, sire genotype and sex of meat rabbits” I rate very highly. The authors correctly motivate the purpose of the undertaken research and the introduction is adequate to the undertaken topic. The description of the results is reliable and the literature presented in the discussion is properly selected. An important element of the manuscript is the assessment of meat quality as a continuation of the research aimed at determining the breeding results of rabbits - hybrids. This is the comprehensiveness of the entire research.

The main remark, for the possible consideration of the authors, is the possibility of performing a statistical analysis using two-factor analysis of variance not for the principal factors but for factor systems, which would allow the use of post-hoc tests to determine the differences between all four experimental groups. Of course, principal factor analysis allows the study of possible factor interactions, and is often used in different researches.

Below I have included a few editorial comments "

Line 64 - correct the order of citation [16, 18, 19]

Line 259 - position 32 appears, and position 32 is only on line 263, change the order. Later, also pos. 32 with pos. 32 should be changed.

Line 411 - correct the order of citation [4, 54]

Line 432 - correct the order of citation [59, 60]

Line 533 and 595 - Bold for the year.

Author Response

At first we would like to thank to the reviewer 3 for the factual comments and proposed corrections to improve the quality of our paper. In the revised manuscript we have taken into consideration all the comments and suggestions.

We performed the post hoc analysis by Tuckey´s test, when the significant interaction between genotype and sex was proved in tables 3-5.  We marked the particular differences among the groups assessed by superscript letters there.

To specific points:

Line 64 - correct the order of citation [16, 18, 19]

We corrected it.

Line 259 - position 32 appears, and position 32 is only on line 263, change the order. Later, also pos. 32 with pos. 32 should be changed.

We revised it.

Line 411 - correct the order of citation [4, 54]

We corrected it.

Line 432 - correct the order of citation [59, 60]

We corrected it.

Line 533 and 595 - Bold for the year.

We corrected it.

Reviewer 4 Report

the article “Proximate chemical composition, amino acids profile and minerals content of meat depending on carcass part, sire genotype 3 and sex of meat rabbits” is well written and well-structured. The double comparison, both on the genotype and on the sex, studies all the possible influences on the quality of the meat in one time. I really appreciated the discussion about the results, which I found to be very through.

For me it can be accepted in the present form. there is only to correct the division into syllables at line n. 57, 70 and 72.

Author Response

At first, we would like to thank to the reviewer 4 for the suggestion to improve the quality of our paper. In the revised manuscript we have taken into consideration it; we corrected the division into syllables at original lines no. 57, 70 and 72, as recommended.

Reviewer 5 Report

The present manuscript describes the effect of genotype and sex on meat composition of rabbits. The effect of genotype and sex on rabbit meat composition has been widely studied in numerous papers; however, the studies on Mecklenburger Schecke sires are less abundant.

In my opinion, this issue has interest from scientific point of view. However, there are some questions that could be convenient clarify. Some results are not properly discussed, mainly significant interactions.

Specific comments:

Abstract

Presenting data in the abstract would improve its value. Rewrite this section according to the following comments in results and discussion sections. Add a sentence with conclusions.

Introduction

The introduction is appropriate.

L41-43: this sentence needs a reference

L52: 2008, recently?

L74-80: The effect of carcass part is missing in the objective of the study.

Material and Methods

Table 1 Please delete units column and put “g/kg” in table title: “Chemical composition (g/kg)…”

L112-114: How much rabbits were used in the experiment, 112 (28 per group) or 96 (24 per group)?

L130-148: Authors should indicate the reference of the method used for amino acid analyses or explain with more detail the method.

L158: SW? Please define the abbreviation

Results

L237-239: these results do not appear in tables, please include it in table 3. it could also be interesting to include growth rate, include carcass weight, carcass yield and leg yield.

Table 3 Please add “carcass traits” to the table title, and include growth rate, carcass weight, carcass yield, leg yield.

Discussion

L264: IMF; Please define the abbreviation

L273-274: Authors cannot ignore the significant genotype x sex interaction observed for crude protein and ash in leg. The authors should do an effort to find and provide an explanation to justify the significant interactions observed. When significant interaction is observed I suggest compare the four groups with a mean comparison test.

L284-285: The explanation is not convincing, the effect of sex is significant, males have more ether extract than females. This result is surprisingly, can you explain it?

L314-323: A significant genotype x sex interaction is observed for total AA and EAA. Again, you should explain it. Again, I suggest a mean comparison test. Idem for L340-370.

Conclusions

Too long for conclusion, it seems a summary of the study. Rewrite taking into account the interactions.

Author Response

At first we would like to thank to the reviewer 5 for the factual comments and proposed corrections to improve the quality of our paper. In the submitted revised manuscript we have taken into consideration all the comments and suggestions.

To specific points:

Abstract

Presenting data in the abstract would improve its value. Rewrite this section according to the following comments in results and discussion sections. Add a sentence with conclusions.

We rewrote the entire abstract respecting changes performed in other revised sections. We added the level of significances for the important results, the sentence with conclusions etc.   

Introduction

L41-43: this sentence needs a reference

We put the appropriate reference for this statement here.

L52: 2008, recently?

We changed it to “earlier”.

 L74-80: The effect of carcass part is missing in the objective of the study.

We added it in the objective of the study.

Material and Methods

Table 1 Please delete units column and put “g/kg” in table title: “Chemical composition (g/kg)…”

We deleted the column “Unit” and put “g/kg” in the title; as recommended also by the reviewer 2.

L112-114: How much rabbits were used in the experiment, 112 (28 per group) or 96 (24 per group)?

There were used 112 weaned rabbits in the beginning of the experiment. At 108 days of age, 48 rabbits were randomly selected (24 rabbits per a genotype; 12 males + 12 females) for meat sampling and following analysis.   

We revised the statement in the current line no. 114.

L130-148: Authors should indicate the reference of the method used for amino acid analyses or explain with more detail the method.

We put the reference of the work Straková et al. (2015), which describes AA analyses used.

L158: SW? Please define the abbreviation.

We explained the short here.

Results

L237-239: these results do not appear in tables, please include it in table 3. It could also be interesting to include growth rate, include carcass weight, carcass yield and leg yield.

Table 3 Please add “carcass traits” to the table title, and include growth rate, carcass weight, carcass yield, leg yield.

As suggested, we included the following traits - average daily gain, carcass weight, carcass dressing and hind legs yield - in the table 3. We revised the title of the table 3.

We revised the results part in light of performed addition of above mentioned carcass traits in tab. 3; currently stated on lines no.172-174.  

Discussion

L264: IMF; Please define the abbreviation.

We explained the short; now on line no. 271.

L273-274: Authors cannot ignore the significant genotype x sex interaction observed for crude protein and ash in leg. The authors should do an effort to find and provide an explanation to justify the significant interactions observed. When significant interaction is observed I suggest compare the four groups with a mean comparison test.

L314-323: A significant genotype x sex interaction is observed for total AA and EAA. Again, you should explain it. Again, I suggest a mean comparison test. Idem for L340-370.

We performed post-hoc testing by Tucke´y test in assessed traits, if the interaction between the sex and genotype was proved to be significant, as recommended. Significant differences among the assessed 4 groups in tab. 3-5 are marked by different superscript letters.  

With respect to the proved interaction effect, we revised discussion parts with willingness to explain revealed findings; specifically on lines no. 283-284, 326-328 and 361-362.

L284-285: The explanation is not convincing, the effect of sex is significant, males have more ether extract than females. This result is surprisingly, can you explain it?

The IMF content of LTL meat was higher in males, which was also observed (p ˃ 0.05) by North et al. (2019) in the LTL of meat rabbit genotypes. It seems that male rabbits may in some cases deposit more IMF into the LTL muscle with advancing age. We revised it on lines no. 293-296.

Conclusions

Too long for conclusion, it seems a summary of the study. Rewrite taking into account the interactions.

 We reworked and shortened it; as recommended it also by the reviewer 1. As suggested, we included in the conclusion the finding about significant effect of interaction between the genotype and sex of rabbits. 

Round 2

Reviewer 5 Report

There has been important improvement in the revised version. The authors have made a great effort to improve the original manuscript. In general, all manuscript sections have been improved; however, some minor revisions are necessary:

L162 Replace “Tuckey” by “Tukey”

Table 2, title: please change “meat quality traits” by “meat composition”

Table 3, title: put “(g/kg of fresh meat)” after “chemical composition”

footnote: please change “water/protein ration” by “water /protein ratio”

Table 4, put “(g/100 g of total crude protein)” after “Amino acid profile” (similar in Table 5)

Table 3, 4 and 5: There are missing superscript letters in some rows, for example crude protein and W/P rows in table 3.

Author Response

We would like to thank to the reviewer 5 for the formal comments to improve the quality of our paper. In the secondly revised manuscript we have taken into consideration all the suggestions again and revised them.

To specific points:

L162 Replace “Tuckey” by “Tukey”

We corrected it.

Table 2, title: please change “meat quality traits” by “meat composition”

We changed it.

Table 3, title: put “(g/kg of fresh meat)” after “chemical composition”

footnote: please change “water/protein ration” by “water /protein ratio”

We revised and changed it.

Table 4, put “(g/100 g of total crude protein)” after “Amino acid profile” (similar in Table 5)

We revised it.

Table 3, 4 and 5: There are missing superscript letters in some rows, for example crude protein and W/P rows in table 3.

We added the missing superscript letters in particular rows of the tab. 3 to 5.